# Education mediating the associations between early life factors and frailty: a cross-sectional study of the UK Biobank

Asri Maharani [ORCID],[1,2] Altug Didikoglu,[3] Terence W O'Neill,[4,5] Neil Pendleton,[3] Maria Mercè Canal,[3] Antony Payton[6]

AM and AD contributed equally.

## ABSTRACT

**Objectives** Exposures in utero and during infancy may impact the development of diseases later in life. They may be linked with development of frailty, although the mechanism is unclear. This study aims to determine the associations between early life risk factors and development of frailty among middle-aged and older adults as well as potential pathways via education, for any observed association.

**Design** A cross-sectional study.

**Settings** This study used data from UK Biobank, a large population-based cohort.

**Participants** 502 489 individuals aged 37–73 years were included in the analysis.

**Primary and secondary outcome measures** Early life factors in this study included being breast fed as a baby, maternal smoking, birth weight, the presence of perinatal diseases, birth month and birth place (in or outside the UK). We developed a frailty index comprising 49 deficits. We used generalised structural equation modelling to examine the associations between early life factors and development of frailty and whether any observed association was mediated via educational attainment.

**Results** A history of breast feeding and normal birth weight were associated with a lower frailty index while maternal smoking, the occurrence of perinatal diseases and birth month with a longer day length were associated with a higher frailty index. Educational level mediated the relationship between these early life factors and frailty index.

**Conclusions** This study highlights that biological and social risk occurring at different stages of life are related to the variations in frailty index in later life and suggests opportunities for prevention across the life course.

## INTRODUCTION

As the world's population ages, a major goal is the attainment of increased life expectancy accompanied by fewer years spent in poor health and with disability and dependency. The worldwide population of older people (65 years and above) is predicted to double from 0.7 billion (9%) in 2019 to 1.5 billion (16%) in 2050.[1] In addition, there is evidence that the number of disability-adjusted life years among those aged 60 years and older is increasing (from 434 million in 1990 to

### STRENGTHS AND LIMITATIONS OF THIS STUDY

⇒ Using a large cohort of British adults in middle and older age, this study was sufficiently powered to identify associations between early life factors and frailty index.
⇒ The questionnaire on early life factors was based on self-report and is therefore subject to recall error.
⇒ The information on the health conditions of the parents is limited, and the data on the breastfeeding duration are not available.
⇒ As the cohort is not nationally representative, the findings cannot be generalised to the general population.

574 million in 2010),[2] which will increase demand for health and care services. As physical disability is an adverse outcome of frailty,[3] more research in geriatrics and gerontology has focused on defining and recognising frailty among older people with the aim of determining preventive and interventional measures.[4]

Frailty can be defined as a state of increased vulnerability resulting from an age-related decline in physiological and cognitive reserves and function following stressor events.[5] The frailty index approach, developed by Rockwood and Mitnitski,[6] measures frailty level as the number of deficits presents over the number of deficits considered, including symptoms, diagnoses, disabilities and functional impairments. Frailty has become more common with the ageing of the population. A systematic review including 240 studies from 62 countries showed that 24% of people aged 50 years and older are frail as calculated using the frailty index approach.[7] Frailty has been found to be associated with adverse health outcomes including loss of mobility, disability, falls, hospitalisation, need for long-term care and death.[8–10] Understanding the factors that are associated with frailty is thus important for developing interventions to prevent frailty

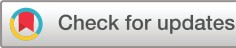

For numbered affiliations see end of article.

**Correspondence to**
Dr Altug Didikoglu;
altug.didikoglu@manchester.ac.uk

and for providing directions for future public health policies.

A growing body of literature acknowledges that the first two decades of human life are critical in determining adult life trajectories. Among the early life factors, body size at birth,[11 12] cigarette smoke exposure in utero,[13] infants exclusively breast fed,[14] birth month[15] and the presence of perinatal diseases[16] have been found to be associated with adult chronic diseases. However, the study linking those factors and frailty is limited. In addition, the evidence on the link between early life factors and occurrence of frailty have been mixed.[17 18] The present study thus aims to determine the associations between early life factors, including a history of being breast fed, maternal smoking, birth weight, the presence of perinatal diseases, birth in or outside the UK and birth month and frailty in UK adults.

Furthermore, this study contributes to the literature investigating the determinants of health in later life by exploring the pathways of early life factors that have a lasting impact on health in middle and old age. The pathway hypothesis posits that early life conditions are important because they are directly associated with late life and because they shape later life experiences,[19 20] including restricted educational attainment and life chances. The most frequently hypothesised pathway between circumstances in early stages of life and adult health is adult socioeconomic status. Pakpahan *et al* showed that socioeconomic factors in adulthood, including education, mediate the link between childhood health and socioeconomic conditions and self-rated health among older Europeans.[19] Because interventions that target common pathways have the potential to reduce frailty, the identification of the pathways of early life factors leading to frailty later in life has substantial public health relevance for the translation of life course epidemiology into practice. The present study considers whether any observed association between early life factors and frailty could be attributed to differences in education attainment (figure 1).

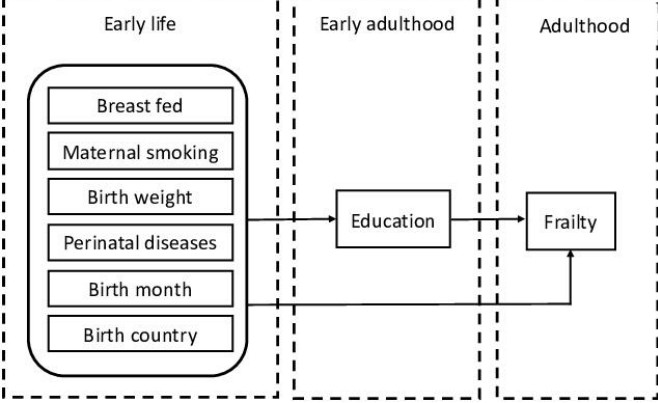

**Figure 1** The pathways of early life factors and impact on frailty among adults.

## METHODS
### Source and sample
Data were drawn from the UK Biobank, a prospective cohort study of the genetic, environmental and lifestyle causes of diseases among adults in the UK.[21] The study involved the collection of extensive questionnaire data and biological samples from, and the performance of, physical examinations of >500 000 respondents enrolled at 22 assessment sites in England, Scotland and Wales between 2006 and 2010. Subjects who took part provided written informed consent for data collection, analysis and linkage; they also completed a touchscreen questionnaire, a nurse-led interview and had their physical measurements taken. The UK Biobank invited adults who were registered with a general practitioner and who lived within reasonable travelling distance of the assessment centre. The current study includes 502 489 individuals aged 37–73 years who had study-specific available data and were not withdrawn from the study.

## MEASURES
### Early life factors
Information by questionnaire was obtained on: maternal smoking in the prenatal and postnatal period, history of being breast fed as a baby, birth month, birth weight, the presence of perinatal diseases and place of birth. We defined maternal smoking based on the question "Did your mother smoke regularly around the time when you were born?" (Data-Field 1787). Respondents were categorised as having been breast fed as babies if they answered 'yes' to the question: "Were you breast fed when you were a baby?" (Data-Field 1677). We retrieved information on birth month from the birth date (Data-Field 52) and treated it as the cosine of the values, representing the rhythmic seasonal length of day and night. We considered this might represent daylight time better than treating it as a categorical variable. This is an approach which we have used in a previous study.[22] Birth months of participants born in the UK and other countries in the southern hemisphere were converted to their antiphase. Information on birth weight was gathered by means of self-reported birth weight in kilograms (Data-Field 20022). We categorised the birth weight into low birth weight (<2500 g), normal birth weight (2500–4000 g) and high birth weight (>4000 g). The presence of perinatal diseases ('ICD-10 chapter XVI: certain conditions originating in the perinatal period') was coded as one based on self-reported medical history (category 2416). We categorised the place birth of the respondents as born in the UK or outside the UK (Data-Field 1647). Answers of 'Do not know' or 'Prefer not to answer' were accepted as missing for all questions.

### Education
The education variable represents the highest educational level completed by the respondents. Qualifications

were categorised as high school or less (reference) and college or university degree (Data-Field 6138).

## Frailty index

Following William *et al*,[23] we derived the frailty index using 49 functional, psychological and social deficits within the range of data variables in the UK Biobank (see online supplemental table 1). We coded the binary variables as 0 or 1, and for ordinal and continuous variables, coding was based on distribution. The total number of deficits was summed and divided by total possible deficits to create a frailty index between 0 and 1, where higher scores indicated greater frailty.

## Covariates

We included demographic and health behaviour as covariates. Demographic information included age (in years; Data-Field 21003), gender (with male as the reference; Data-Field 31) and ethnicity (other than Caucasian as the reference or Caucasian; Data-Field 21000). Health behaviours included physical activity, alcohol intake and smoking status. Physical activity was measured as the number of days per week respondents engaged in at least 10 min of moderate or vigorous physical activity (Data-Field 884, Data-Field 904). Respondents were classified as non-current smokers (reference) or current smokers (Data-Field 20116). Alcohol intake status was classified as non-current (reference) or current alcohol drinking (Data-Field 20117).

## Statistical analyses

Descriptive statistics were used to summarise subject characteristics including means and SD for continuous variables and frequencies and percentages for categorical variables. We looked at the associations between frailty index and both early life factors and other covariates using unpaired t-tests (dichotomous variables), analysis of variance (categorical variables) and Pearson's correlation (continuous variables).

We first performed a multivariate regression model including early life factors, education and covariates (age, gender, ethnicity, smoking, alcohol drinking, physical activity). We further handled missing data using multivariate imputation by chained equations[24] (using Stata's mi program).[25] Twenty imputations were used.

The structural equation model (SEM) has been widely used to investigate complex relationships between variables in epidemiological studies.[26] SEM can be used to resolve the endogeneity problem between variables and to explore direct, indirect and total effects between exogenous and endogenous variables. It can jointly test a variety of hypotheses that involve different types of complicated cause-effect relationships. However, all responses are assumed to be continuous, even when a variable is binary or categorical. In our analysis we include binary (education). To address this, we used a generalised structural equation model (GSEM) to identify the link between early life factors and frailty index

and the mediating effect of education on that relationship. A GSEM combines generalised linear model (GLM) estimation and SEM modelling estimation; it can accommodate binary, ordinal, counted and categorical data.[27] Using maximum likelihood estimators, GLM estimators are based on a density function, allowing the direct use of all types of data.[28] The analyses were performed using MPlus V.8. We examined education as mediators of the relationship in the GSEM model, which were controlled for age, gender, ethnicity and health behaviours (model fit information: $\chi^2$=5049.35, p=0.00; RMSEA (root mean square error of approximation)=0.06, CFI (comparative fit index)=0.82; WRMR (weighted root mean square residual)=13.01).

## Patient and public involvement

Patients and/or the public were not involved in the design, or conduct, or reporting, or dissemination plans of this research.

# RESULTS

## Subjects

The study sample consisted of 502 489 respondents with an average age of 56.53 years (SD=8.10 years) (table 1). Just under half (45%) of the respondents were male, and most were Caucasian (94.59%). Around one-third of the respondents had graduated from college or university. The proportion of respondents whose mothers smoked regularly around the time of their birth was 29%. More than 72% of respondents were breast fed as babies, and 0.18% had perinatal diseases. Ten per cent of respondents had low birth weight, while 13% of them had high birth weight; 91% of the respondents were born in the UK. Just over two-thirds of subjects reported engaging in at least 10 min of moderate or vigorous physical activity at least 3 days per week; 92% consumed alcohol and 11% were current smokers.

## Early life factors, covariates and frailty index

In bivariate analyses, compared with those whose mothers did not smoke around birth, maternal prenatal and postnatal smoking was associated with a significantly higher frailty index (0.146 vs 0.133) as was the presence of perinatal diseases (0.149 vs 0.138) and being born in the UK (0.138 vs 0.137). A history of breast feeding was associated with a lower frailty index (0.134 vs 0.137). Low (0.149 vs 0.131) and high (0.136 vs 0.131) birth weight were associated with higher frailty scores compared with normal birth weight. Shorter daylight hours at birth (r=−0.01) were associated with lower frailty indices. As expected, the frailty index was higher among women than among men and in those with lower educational attainment. The frailty index was also higher in smokers, non-drinkers and those who engaged in less physical activity.

In regression analyses, the effects of early life factors and covariates on the frailty index appeared similar in terms of both magnitude and direction when using

**Table 1** Subject characteristics (n=502 489)

| Variable | Percentage or mean (SD)* | Mean (SD) of frailty index† | Bivariate association with frailty index‡ |
|---|---|---|---|
| Frailty index, mean (SD) | 0.14 (0.08) | | |
| *Early life factors* | | | |
| Maternal smoking around birth, % | | | P<0.0001 |
| No | 70.75% | 0.133 (0.073) | |
| Yes | 29.25% | 0.146 (0.078) | |
| Breast fed as a baby, % | | | P<0.0001 |
| No | 27.65% | 0.137 (0.076) | |
| Yes | 72.35% | 0.134 (0.074) | |
| Birth weight, % | | | P<0.0001 |
| Low birth weight | 10.26% | 0.149 (0.080) | |
| Normal birth weight | 76.34% | 0.131 (0.073) | |
| High birth weight | 13.40% | 0.136 (0.076) | |
| Birth month, % | | | P=0.0002 |
| January | 8.44% | 0.138 (0.076) | |
| February | 7.96% | 0.137 (0.075) | |
| March | 8.98% | 0.138 (0.075) | |
| April | 8.59% | 0.139 (0.076) | |
| May | 8.98% | 0.138 (0.076) | |
| June | 8.45% | 0.139 (0.076) | |
| July | 8.48% | 0.139 (0.076) | |
| August | 8.24% | 0.138 (0.076) | |
| September | 8.14% | 0.138 (0.075) | |
| October | 8.06% | 0.137 (0.076) | |
| November | 7.63% | 0.137 (0.075) | |
| December | 8.03% | 0.138 (0.076) | |
| Perinatal diseases, % | | | P<0.0001 |
| No | 99.82% | 0.138 (0.075) | |
| Yes | 0.18% | 0.149 (0.084) | |
| Born in the UK, % | | | P=0.0381 |
| No | 8.96% | 0.137 (0.076) | |
| Yes | 91.04% | 0.138 (0.075) | |
| *Sociodemographics* | | | |
| Age (years), mean (SD) | 56.53 (8.10) | | r=0.16, p<0.0001 |
| Gender, % | | | P<0.0001 |
| Female | 54.40% | 0.141 (0.075) | |
| Male | 45.60% | 0.134 (0.076) | |
| Ethnicity, % | | | P<0.0001 |
| Other | 5.41% | 0.141 (0.078) | |
| Caucasian | 94.59% | 0.138 (0.075) | |
| Education, % | | | P<0.0001 |
| Less than college | 67.27% | 0.145 (0.077) | |
| College or university degree | 32.73% | 0.122 (0.069) | |
| *Health behaviours* | | | |
| Moderate or vigorous physical activity, % | | | P<0.0001 |
| None | 10.75% | 0.160 (0.085) | |

Continued

**Table 1** Continued

| Variable | Percentage or mean (SD)* | Mean (SD) of frailty index† | Bivariate association with frailty index‡ |
|---|---|---|---|
| 1 day | 7.11% | 0.134 (0.072) | |
| 2 days | 13.40% | 0.133 (0.072) | |
| 3 days or more | 68.75% | 0.135 (0.073) | |
| Current alcohol consumption, % | | | P<0.0001 |
| No | 8.08% | 0.166 (0.088) | |
| Yes | 91.92% | 0.135 (0.074) | |
| Current smoking, % | | | P<0.0001 |
| No | 89.39% | 0.135 (0.074) | |
| Yes | 10.61% | 0.159 (0.084) | |

*Presented are means (SD) for continuous variables and percentages for categorical variables. The maternal smoking variable includes 13.86% missing data, the breast fed as a baby variable includes 23.64% missing data, the birth weight variable includes 44.88% missing data, the education variable includes 2.02% missing data and the moderate or vigorous physical activity variable includes 2.43% missing data.
†Presented are the means (SD) of the frailty index per group.
‡Bivariate analyses are unpaired t-tests for binary variables, analysis of variance for ordinal variables and Pearson's correlation for continuous variables.

both non-imputed and imputed data (see online supplemental table 2). In these multivariate regression analyses, adjusting for age, gender and health behaviours, birth month with longer hours of daylight, having a low and high birth weight, maternal smoking, not being breast fed as baby, perinatal diseases and born in the UK had positive and significant associations with frailty index.

### Mediation analysis

In the GSEM model, education mediated the association between early life factors and frailty index among middle-aged and older adults, supporting the pathway hypothesis. Table 2 presents the total, direct and indirect effects for each of the early life factors on the frailty index. Maternal smoking (direct effect: coefficient=0.068, z=33.40; indirect effect: coefficient=0.011, z=25.54) and low birth weight (direct effect: coefficient=0.041, z=20.93; indirect effect: coefficient=0.003, z=9.18) and high birth weight (direct effect: coefficient=0.013, z=6.34; indirect effect: coefficient=0.001, z=4.09) directly and indirectly affected

the frailty index compared with normal birth weight. The direct and indirect effects of being breast fed as a baby on having a lower frailty index were −0.022 (z=−10.36) and −0.009 (z=−22.91). Perinatal diseases had significant direct effect on higher frailty index (coefficient=0.007, z=3.83), but it had no indirect effect on the frailty index (coefficient=0.000, z=0.27). Being born in the UK, differently, had a significant indirect effect on higher frailty index (coefficient=0.016, z=31.24), but it had no direct effect on the frailty index (coefficient=0.002, z=0.74). Birth months with a short daylength were directly (coefficient=−0.006, z=−2.91) and indirectly (coefficient=−0.001, p=−2.35) associated with lower frailty scores.

Education mediated the links between early life factors and frailty index (figure 2). Participants born in the UK had a lower probability of completing higher education (coefficient=−0.130, z=−44.65). Having been breast fed as a baby (coefficient=0.076, z=26.87) was associated with higher educational attainment, while maternal smoking

**Table 2** Total, direct and indirect effects of early life factors on frailty index

| | Total effects | Direct effects | Indirect effects |
|---|---|---|---|
| Breast fed as a baby | −0.031 (0.002)* | −0.022 (0.002)* | −0.009 (0.000)* |
| Maternal smoking around birth | 0.079 (0.002)* | 0.068 (0.002)* | 0.011 (0.000)* |
| Low birth weight | 0.045 (0.002)* | 0.041 (0.002)* | 0.003 (0.000)* |
| High birth weight | 0.015 (0.002)* | 0.013 (0.002)* | 0.001 (0.000)* |
| Birth month (cos) | −0.007 (0.002)† | −0.006 (0.002)† | −0.001 (0.000)† |
| Perinatal diseases | 0.007 (0.002)* | 0.007 (0.002)* | 0.000 (0.000) |
| Born in the UK | 0.018 (0.002)* | 0.002 (0.002) | 0.016 (0.001)* |

*p<0.001
†p<0.005

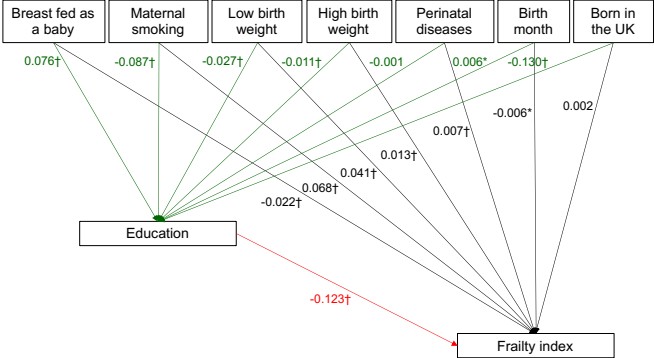

**Figure 2** Generalised structural equation models to identify the association between early life factors and frailty index, and education as mediators of the relationship between early life factors and frailty index. *p<0.05; †p<0.001.

was associated with lower educational attainment (coefficient=−0.087, z=−31.41). Both low (coefficient=−0.027, z=−9.39) and high birth weight (coefficient=−0.011, z=−4.11) was related to lower education attainment compared with normal birth weight. Birth months with short daylight was related to higher education with the lowest effect size (coefficient=0.006, z=2.35). Higher education was also associated with a lower frailty index (coefficient=−0.123, z=−44.30). Among covariates with greater effect sizes, older age (coefficient=0.178, z=83.25), lower activity levels (coefficient=−0.088, z=−45.61) and smoking (coefficient=0.106, z=56.36) were associated with a higher frailty index. Drinking alcohol is related to lower frailty index (coefficient=−0.093; z=−51.63).

## DISCUSSION

Using data from UK Biobank, we found that a history of breast feeding was associated with a lower frailty index, while maternal smoking, having low or high birth weight, perinatal diseases and birth month with longer day length were associated with a higher frailty index. This study provides the first evidence that educational attainment level mediates the association between early life factors and frailty index.

Early life factors have previously been linked with higher frailty and chronic disease risk later in life.[29][30] Our findings highlight the importance of early life factors in determining frailty in middle age and older individuals. Maternal smoking was directly associated with higher frailty compared with those who were not exposed to maternal smoking. Evidence has suggested that cigarette smoke exposure in utero is linked to the development of chronic diseases later in life, including type 2 diabetes, obesity, certain cancers and respiratory disorders.[13] We also showed that this association was mediated by educational attainment. This is in line with a previous study which reported lower academic achievements of adolescents whose mothers smoked during pregnancy.[31] Maternal smoking during pregnancy was also found to be correlated with the children's cognitive function.[32]

There is some evidence of a link between early life factors and occurrence of frailty. In a recent study in Finland, greater weight, length and body mass index at birth were associated with a lower risk of frailty later in life.[17] In our study, having low or high birth weight were associated with higher frailty index compared with having a normal birth weight, both directly and indirectly through education. Bleker *et al* found that prenatal undernutrition was not associated with frailty but was associated with poorer health in old age, including slower gait speed and lower physical functioning which are components of the frailty phenotype, and the findings remained significant after inclusion of an extensive set of control variables including adult socioeconomic status.[18] Low birth weight is associated with increased risk of age-related diseases in prior review, and insulin-like growth factor 1 is the key driver of this process.[33] High birth weight may be the results of maternal obesity[34] and a study in Finland found that being born large for gestational age at term was associated with thicker carotid intima medial as the marker of subclinical atherosclerosis.[35] We also found that individuals who reported that they were breast fed have a lower frailty score. Infants exclusively breast fed have been found to have a lower risk of obesity, type 2 diabetes and high blood pressure in adulthood.[14]

Birth month is associated with lower frailty index scores with a limited effect size in our study. In a large study in the USA with 1 749 400 individuals showed that spring summer-born individuals have a relatively higher cardiovascular disease risk than autumn-winter born individuals and these seasons coincide with lower life expectancy.[36] This study showed that cardiovascular diseases and several chronic diseases were associated with season of birth, having a different seasonal pattern. The underlying mechanisms may differ for each of these associations, such as sensitisation to allergens or vitamin D deficiency.[36] Another possible mechanism is that differential light exposure during perinatal period influences development of the biological clock, in turn influencing later-life circadian rhythms and the sleep system, which are essential for health.[37] In European countries, it was shown that spring/summer born participants compared with autumn had higher frailty scores but this effect seemed independent of education.[38] However, we found an indirect effect of season of birth through education. The indirect relationship of season of birth and frailty may be due to social factors such as the UK September date cut-off for starting education, which is in line with our findings showing an association between winter-born individuals and higher educational attainment.[39][40]

Our results further suggest that having a perinatal disease was associated directly with higher frailty index scores. This finding is in keeping with that newborns' perinatal complications are related to accelerated ageing at midlife.[16] Being born in the UK affected the frailty index indirectly through education, but not directly. Respondents who were not born in the UK were likely to have higher education attainment, which may enable

better maintenance of health during older ages. However, we should note that our sample in this analysis may not be representative of the general population, and that participants were categorised as being born outside the UK without taking into account the country of origin and their socioeconomic background. In our analysis, we observed that education levels mediate the link between the other early life factors and the frailty index. Early life factors have a significant relationship with educational attainment, and higher education attainment is linked to a lower frailty index. This result is broadly in keeping with a prior study in Sweden which found that the associations between childhood conditions and various old age health indicators (musculoskeletal disorders, cardiovascular disease, self-rated health and impaired mobility) are mediated by education.[41] Prior research on the biological and psychological pathways linked childhood health and socioeconomic conditions to self-reported health status among older adults in 15 European countries.[19] Prior studies have shown that the life-course trajectories of socioeconomic attainment could be altered by physical and social conditions,[42] and both childhood and adult conditions may impact health decades later.[43] Our findings have potential implications for policies aiming at preventing frailty among older adults. Subsequent circumstances mediate the impact of early life factors on frailty later in life, and our study suggests that interventions such as improving education in midlife may mitigate early life disadvantages.

Our findings are based on a large and well-characterised cohort. There are, however, a number of limitations to be consider in interpreting the results. First, information concerning early life factors in this study was based on self-report and is therefore subject to recall error. The likely effect of such error would be to underestimate the relationship between these factors and the frailty index. Second, we have limited access to the health conditions of the parents. A broad range of conditions which are comprised in the frailty index bear a hereditary risk, thus taking into account the health conditions of the parents is important in assessing the independent associations with frailty. Future studies may include the health conditions of the parents as the covariates. Third, the information on breastfeeding duration is unavailable. Breast feeding for weeks rather than months may confer different outcomes. A dose-response relationship thus cannot be assessed. Finally, these data were based on a sample of predominantly Caucasian men and women and should be extrapolated beyond this group with caution.[44]

In conclusion, this study indicates an association between early life factors and frailty later in life. Early life conditions are important as the start of a mediated, incremental process during the life course. A comprehensive understanding of the determinants of frailty among middle-aged and older adults requires attention to exposures throughout the entire life course, with a special focus on the in utero and infancy stages and the chains of associated socioeconomic conditions that connect over the life course. Applying a life-course perspective to health in adulthood and old age should have implications for public health interventions, social policy and further research. Early life is not the only period for any potential successful intervention; as our findings show, early life disadvantages may be offset by education. Interventions throughout the life course, and especially during early life, could substantially reduce the health burden later in life.

**Author affiliations**
[1]Division of Population Health, Health Services Research & Primary Care, The University of Manchester, Manchester, UK
[2]Department of Nursing, Manchester Metropolitan University, Manchester, UK
[3]Division of Neuroscience & Experimental Psychology, The University of Manchester, Manchester, UK
[4]Centre for Epidemiology Versus Arthritis, Division of Musculoskeletal & Dermatological Sciences, The University of Manchester, Manchester, UK
[5]UK & NIHR Biomedical Research Centre, Manchester University NHS Foundation Trust, Manchester, UK
[6]Division of Informatics, Imaging & Data Sciences, The University of Manchester, Manchester, UK

**Contributors** AD and AM performed the data analysis. AD and AM drafted the manuscript. NP, TWO'N, MMC and AP were involved in planning and supervised the work. All authors discussed the results and commented on the manuscript. AM is responsible for the overall content as the guarantor.

**Funding** AD was supported by a grant from the Republic of Turkey Ministry of National Education.

**Competing interests** None declared.

**Patient and public involvement** Patients and/or the public were not involved in the design, or conduct, or reporting, or dissemination plans of this research.

**Patient consent for publication** Not applicable.

**Ethics approval** This study was conducted as part of UK Biobank Project Number 41877 and is covered by the generic ethics approval for UK Biobank studies from the NHS National Research Ethics Service (16/NW/0274). Participants gave informed consent to participate in the study before taking part.

**Provenance and peer review** Not commissioned; externally peer reviewed.

**Data availability statement** Data may be obtained from a third party and are not publicly available. Data are available in a public, open access repository. Other researchers can apply for UK Biobank data to answer specific research questions.

**ORCID iD**
Asri Maharani http://orcid.org/0000-0002-5931-8692

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
