## [Reviewer comments · BMJ Open]

ARTICLE DETAILS

TITLE (PROVISIONAL)	Education mediating the associations between early life factors and frailty: a cross-sectional study of the UK Biobank
AUTHORS	Maharani, Asri; Didikoglu, Altug; O'Neill, Terence; Pendleton, Neil; Canal, Maria; Payton, Antony

VERSION 1 – REVIEW

REVIEWER	Lewis, Emma G Newcastle University, Population Health Sciences Institute
REVIEW RETURNED	26-Oct-2021

GENERAL COMMENTS	Thank you for the opportunity to review this very interesting study linking early life events with mid- and later-life frailty. It is well written and structured. Here are a few comments for your consideration: There are limitations in your FI as cognitive impairment has not been included and there are no measures of self-reported or objectively measured functioning e.g. basic and instrumental activities of daily living from washing/dressing to preparing meals and paying bills. A few of the deficits included are not clearly age-associated e.g. hayfever, eczema and asthma. These limitations should be acknowledged at least, but if possible more functional and cognitive deficits should be included in the FI. The finding of an association with birth month is interesting, especially given previous conflicting findings. Would it be possible to discuss possible mechanisms/theories relating to birth month? Is it associated with education and the academic school year for example, thus does it add to your finding that education and income are mediators of these findings. It is a limitation that breast feeding duration is unknown, breast feeding for weeks rather than months may confer different outcomes, a dose response relationship cannot be assessed. Please mention this limitation briefly. I have attached a draft with a few small editing suggestions – contact publisher to view.
---

REVIEWER	Leung, Vania University of Illinois at Chicago
REVIEW RETURNED	27-Nov-2021

GENERAL COMMENTS	This was a very interesting paper that looked to examine potential risk factors early in life that could be associated with frailty, more specifically looking at maternal smoking, breastfeeding, birth months/days, and birth weight. The results were a bit morbid from a clinical standpoint implying that many early life factors appear to mediate an association with frailty. Though clinically speaking, it may
---

	urge future clinical recommendations to push for more support for patients to obtain these potentially life-changing early life risk factors. The frailty index used is not one of the standard measurements of frailty (eg Fried's) but is one created for the UKB cohort based on accumulation of deficits as described by Rookwood et al. Limitations of study were the recall error that a lot of the early life factors were based on and a very large Caucasian population - which the authors have mentioned at the end of the paper. I'm not as familiar with the way the data was presented in the mediation analysis (the coefficients and z-scores), and recommend asking a statistician to review.
--	---

REVIEWER	Aminu, Abodunrin Robert Gordon University
REVIEW RETURNED	29-Nov-2021

GENERAL COMMENTS	This is a nice piece of work and an interesting article to read. It has the potential to broaden the discussion of frailty prevention. You may need to clarify more around the use of birth weight as a continuous variable as this should normally have an acceptable range. It is well known that higher birth weight (e.g in gestational diabetes) is linked to higher risk of future diseases. Again you reported that early factor (birth from non-drinkers) was associated with higher risk of frailty from your result but you have not explained this further. Overall, well done! Your introductory section may require so re-written as the later part of it almost seem like a discussion of already found outcome. One to note is the use of birth weight as a continuous variable because of the known association between high birth weight (common with maternal diseases like gestational diabetes) and risk of adult diseases. The risk of frailty is expectedly higher among those with multiple disease conditions, a major consideration when using the Frailty Index, so where does your result fit. You may decide to use cut-offs for the birth weight or explain the implication of using it in a continuous form in your discussion. Result on page 13 showed that long daylight hours was negatively correlated with FI scores (coef. =-0.0007, z=-3.07) but else where you mentioned that fewer hours of daylight were associated with a significantly lower frailty index (?). Does this have to do with the way the variable is measured or just contradictory. Again, you reported that early factor (birth from non-drinkers) was associated with higher risk of frailty, but you have not mentioned this in your discussion. Previous work (e.g Kojima 2018) suggest otherwise, can you compare?
--

REVIEWER	wu, chen kai Duke Kunshan University, Global Health Research Center
REVIEW RETURNED	08-Feb-2022

GENERAL COMMENTS	The use of the generalized SEM is appropriate. However, the absence of goodness of fit indices in Stata does not mean these important indices can't be computed theoretically. I believe Mplus, which is arguably the best statistical software package for conducting SEM, could report these indices. I would strongly
---

	encourage the authors to repeat their analyses in Mplus. This choice of using persons who were at least 60 years of age in a sensitivity analysis seems arbitrary. More justifications are need. The sample size of the UK Biobank is huge, which is certainly a plus for conducting research. However, a super large sample size could lead to many statistically significant but clinically insignificant results. In other words, even a tiny difference or a very weak association would be detected by such a large sample size. For example, the authors found that a history of breastfeeding was associated with a lower frailty index, while the difference was extremely tiny (0.134 vs. 0.137). I would encourage the authors to present effect size for each important association and interpret the results carefully. The conclusion that the direct effect of early life risk factors on frailty was mediated by education and income is valid from a statistical perspective. However, the actual difference was tiny for all six early life risk factors. For example, the direct effect changed from -0.005 to -0.003 for breastfeeding after including education and income in the model. I would suggest that the authors present standardized coefficients as well to ease the comparisons between different regression coefficients. The authors also need to report the direct effect, indirect effect, and total effect for each early life risk factor.
--	--

VERSION 1 – AUTHOR RESPONSE

Comments from Reviewer #1

Thank you for the opportunity to review this very interesting study linking early life events with mid- and later-life frailty.

It is well written and structured.

Here are a few comments for your consideration:

There are limitations in your FI as cognitive impairment has not been included and there are no measures of self-reported or objectively measured functioning e.g. basic and instrumental activities of daily living from washing/dressing to preparing meals and paying bills. A few of the deficits included are not clearly age-associated e.g. hayfever, eczema and asthma. These limitations should be acknowledged at least, but if possible more functional and cognitive deficits should be included in the FI.

Authors' response

We thank you for this comment and suggestion. We constructed the Frailty Index based on the frailty indices developed specifically for the UK Biobank participants (Williams et al., 2017). This FI has been validated. We thus avoid to create new frailty indices which will need to be validated first before being used in the analysis.

Reference:

Williams DM, Jylhävä J, Pedersen NL, Hägg S. A frailty index for UK Biobank participants. *J Gerontol A Biol Sci Med Sci.* 2019;74:582-587.

Comments from Reviewer #1

The finding of an association with birth month is interesting, especially given previous conflicting findings. Would it be possible to discuss possible mechanisms/theories relating to birth month? Is it

associated with education and the academic school year for example, thus does it add to your finding that education and income are mediators of these findings.

Authors' response

Thank you for this comment. We have discussed possible mechanisms related to season of birth effect on health and frailty such as vitamin D, development of sleep system and early exposure to allergens. Season of birth effect was mediated by education in our analyses. This, indeed, may be related to academic school year with winter-born individuals having higher education levels. We have added this discussion in our revisions. We should note that season of birth effect had the lowest effect size both directly and indirectly in our analyses. We added this limitation clearly in our revisions.

Comments from Reviewer #1

It is a limitation that breast feeding duration is unknown, breast feeding for weeks rather than months may confer different outcomes, a dose response relationship cannot be assessed. Please mention this limitation briefly.

Authors' response

Thank you for drawing our attention to this important issue. We have included this point in the limitations in the Discussion section.

Comments from Reviewer #1

I have attached a draft with a few small editing suggestions.

****Please also see attached comments: 'bmjopen-2021-057511_Proof_hi.pdf****

Authors' response

Thank you for the detail input. We have revised the manuscript based on the comments in 'bmjopen-2021-057511_Proof_hi.pdf'.

Comments from Reviewer #2

This was a very interesting paper that looked to examine potential risk factors early in life that could be associated with frailty, more specifically looking at maternal smoking, breastfeeding, birth months/days, and birth weight. The results were a bit morbid from a clinical standpoint implying that many early life factors appear to mediate an association with frailty. Though clinically speaking, it may urge future clinical recommendations to push for more support for patients to obtain these potentially life-changing early life risk factors. The frailty index used is not one of the standard measurements of frailty (eg Fried's) but is one created for the UKB cohort based on accumulation of deficits as described by Rookwood et al. Limitations of study were the recall error that a lot of the early life factors were based on and a very large Caucasian population - which the authors have mentioned at the end of the paper. I'm not as familiar with the way the data was presented in the mediation analysis (the coefficients and z-scores), and recommend asking a statistician to review.

Authors' response

Thank you for the positive feedback. We agree with the points on the feedback. As we have stated above, we constructed the Frailty Index based on the frailty indices developed specifically for the UK Biobank participants (Williams et al., 2017) and this FI has been validated.

Comments from Reviewer #3

This is a nice piece of work and an interesting article to read. It has the potential to broaden the discussion of frailty prevention.

You may need to clarify more around the use of birth weight as a continuous variable as this should

normally have an acceptable range. It is well known that higher birth weight (e.g in gestational diabetes) is linked to higher risk of future diseases.

Authors' response

Thank you for the supportive feedback. We have revised the analyses and categorised the birth weight into low birth weight (<2,500 g), normal birth weight (2,500 – 4,000), and high birth weight (>4,000 gr) following the study by Gill et al. (2013).

Reference:

Gill, S. V., May-Benson, T. A., Teasdale, A., & Munsell, E. G. (2013). Birth and developmental correlates of birth weight in a sample of children with potential sensory processing disorder. *BMC Pediatrics*, 13, 29. <https://doi.org/10.1186/1471-2431-13-29>

Comments from Reviewer #3

Again you reported that early factor (birth from non-drinkers) was associated with higher risk of frailty from your result but you have not explained this further.

Authors' response

Apologies if this is not clear. We did not use alcohol drinking of mothers as an early life factor. We did not have such variable. We have adulthood drinking behaviour and we used it as a covariate. We try to revise our method and results to make it clear.

Comments from Reviewer #3

You can see the attached document for my feedback.
Overall, well done!

**Please also see attached comments: 'Review feedback_v1.docx'

Authors' response

Thank you for the detail feedback. We have revised the manuscript based on them (please see below) explanation for each point from 'Review feedback_v1.docx'

Comments from Reviewer #3

Your introductory section may require so re-written as the later part of it almost seem like a discussion of already found outcome.

Authors' response

We have rewritten the introductory section.

Comments from Reviewer #3

One to note is the use of birth weight as a continuous variable because of the known association between high birth weight (common with maternal diseases like gestational diabetes) and risk of adult diseases. The risk of frailty is expectedly higher among those with multiple disease conditions, a major consideration when using the Frailty Index, so where does your result fit. You may decide to use cut-offs for the birth weight or explain the implication of using it in a continuous form in your discussion.

Authors' response

We have revised the analyses and categorised the birth weight into low birth weight (<2,500 g), normal birth weight (2,500 – 4,000), and high birth weight (>4,000 gr) following the study by Gill et al. (2013).

Comments from Reviewer #3

Result on page 13 showed that long daylight hours was negatively correlated with FI scores (coef. = -0.0007, $z = -3.07$) but elsewhere you mentioned that fewer hours of daylight were associated with a significantly lower frailty index (?). Does this have to do with the way the variable is measured or just contradictory.

Authors' response

This is because of the way the variable is measured. We used cosine of birth month as a continuous variable representing the length of nights (December is 1, June is -1).

Comments from Reviewer #3

Again, you reported that early factor (birth from non-drinkers) was associated with higher risk of frailty, but you have not mentioned this in your discussion. Previous work (e.g. Kojima 2018) suggest otherwise, can you compare?

Authors' response

Apologies if this is not clear. We did not use alcohol drinking of mothers as an early life factor. We did not have such variable. We have adulthood drinking behaviour and we used it as a covariate. We try to revise our method and results to make it clear.

Comments from Reviewer #4

The use of the generalized SEM is appropriate. However, the absence of goodness of fit indices in Stata does not mean these important indices can't be computed theoretically. I believe Mplus, which is arguably the best statistical software package for conducting SEM, could report these indices. I would strongly encourage the authors to repeat their analyses in Mplus.

Authors' response

Thank you for the comment. We have re-done our analyses in MPlus, and provided the goodness of fit indices. After we did the analyses in MPlus, the goodness of fit indices is very low for the model. We thus removed income from the model to improve the goodness of fit indices.

Comments from Reviewer #4

This choice of using persons who were at least 60 years of age in a sensitivity analysis seems arbitrary. More justifications are need.

Authors' response

We have omitted the sensitivity analysis including only those 60 years and older.

Comments from Reviewer #4

The sample size of the UK Biobank is huge, which is certainly a plus for conducting research. However, a super large sample size could lead to many statistically significant but clinically insignificant results. In other words, even a tiny difference or a very weak association would be detected by such a large sample size. For example, the authors found that a history of breastfeeding was associated with a lower frailty index, while the difference was extremely tiny (0.134 vs. 0.137). I would encourage the authors to present effect size for each important association and interpret the results carefully.

Authors' response

We have presented the standardised effect as the output of MPlus to show both direct and indirect effect of each early life factors.

Comments from Reviewer #4

The conclusion that the direct effect of early life risk factors on frailty was mediated by education and

income is valid from a statistical perspective. However, the actual difference was tiny for all six early life risk factors. For example, the direct effect changed from -0.005 to -0.003 for breastfeeding after including education and income in the model.

Authors' response

We have presented the standardised effect as the output of MPlus to show both direct and indirect effect.

Comments from Reviewer #4

I would suggest that the authors present standardized coefficients as well to ease the comparisons between different regression coefficients.

Authors' response

We have presented the standardized coefficients to ease the comparisons between different regression coefficients.

Comments from Reviewer #4

The authors also need to report the direct effect, indirect effect, and total effect for each early life risk factor.

Authors' response

We have included the direct effect, indirect effect and the total effect for each early life risk factor in the revised document.

VERSION 2 – REVIEW

REVIEWER	Lewis, Emma G Newcastle University, Population Health Sciences Institute
REVIEW RETURNED	11-Jul-2022

GENERAL COMMENTS	The authors have addressed all the comments raised by the reviewers. The new additional methods and results should be checked for the statistics and language. I have attached an edited version with suggestions to improve the clarity – contact publisher to view.
---

REVIEWER	Leung, Vania University of Illinois at Chicago
REVIEW RETURNED	03-Oct-2022

GENERAL COMMENTS	No further comments.
----------------------

REVIEWER	wu, chen kai Duke Kunshan University, Global Health Research Center
REVIEW RETURNED	03-Oct-2022

GENERAL COMMENTS	No further comments.
----------------------

VERSION 2 – AUTHOR RESPONSE

Comments from Editor

The authors have addressed all the comments raised by the reviewers. The new additional methods and results should be checked for the statistics and language. I have attached an edited version with suggestions to improve the clarity.

Authors' response

Thank you for the detailed comments. We have revised the manuscripts based on the in-text comments provided. Some points are:

- We have re-write the sentence in Results section (page 11, last paragraph) to make it clearer into "Birth months with a short daylight were directly (coef.=-0.006, z=-2.91) and indirectly (coef.=-0.001, p-value=-2.35) associated with lower frailty scores.
- We have added the information of the study site in Discussion section (page 13, paragraph 1), and revised the sentence into: "In our study, having low or high birth weight were associated with higher frailty index compared to having a normal birthweight, both directly and indirectly through education."
- We have mentioned in Discussion section (page 13, paragraph 1) that the two factors in the study by Bleker et al. are components of the frailty phenotype, another method for measuring frailty, and revised the sentence into: "Bleker and colleagues found that prenatal undernutrition was not associated with frailty but was associated with poorer health in old age, including slower gait speed and lower physical functioning which are components of the frailty phenotype, and the findings remained significant after inclusion of an extensive set of control variables including adult socioeconomic status [18]."
- We have added the information in our sentence regarding the US study in the Discussion section (page 13, paragraph 2), and revised the sentence into: "In a large study in the US with 1,749,400 individuals showed that spring summer-born individuals have a relatively higher cardiovascular disease risk than autumn-winter born individuals and these seasons coincide with lower life expectancy [36]."
- We have corrected the in-text citing style in the Discussion section (page 14, paragraph 2).
- We have revised the sentence in Discussion section (page 15, paragraph 1) into "Prior studies have shown that the life-course trajectories of socioeconomic attainment could be altered by physical and social conditions [42], and both childhood and adult conditions may impact health decades later [43]."